# Hemorrhagic Transformation after Intravenous Tissue Plasminogen Activator Administration in Acute Distal Middle Cerebral Artery Occlusion

**DOI:** 10.3390/diagnostics12020398

**Published:** 2022-02-03

**Authors:** Chan-Hyuk Lee, Sang Hak Yi, Byoung-Soo Shin, Hyun Goo Kang

**Affiliations:** 1Department of Neurology, Jeonbuk National University Medical School and Hospital, Jeonju 54907, Korea; bluewave0210@gmail.com (C.-H.L.); sbsoo@jbnu.ac.kr (B.-S.S.); 2Department of Neurology, Wonkwang University School of Medicine, Iksan 54538, Korea; md0626@naver.com; 3Research Institute of Clinical Medicine of Jeonbuk National University—Biomedical Research Institute of Jeonbuk National University Hospital, Jeonju 54907, Korea

**Keywords:** atrial fibrillation, cerebral emboli, reperfusion injury

## Abstract

Atrial fibrillation and cerebral embolism are known to increase the risk of hemorrhagic transformation (HT). In addition, a sufficient number of collateral vessels in acute ischemic stroke can maintain the ischemic penumbra and prevent progression to the ischemic core, while an insufficient number of collateral vessels increase the HT risk after therapeutic recanalization. In this case, when the middle cerebral artery is recanalized, reperfusion injury may occur in the basal ganglia due to insufficient collateral vessels.

A 68-year-old man was admitted to the emergency room with left hemiparesis, asomatognosia, and dysarthria within 1 h of symptoms onset. Atrial fibrillation was confirmed based on the electrocardiogram test performed at the time of admission. Brain computed tomography (CT) angiography showed occlusion of the distal portion of the right middle cerebral artery (MCA) (Figure 1A). CT perfusion images showed perfusion mismatch in the right MCA territory, and intravenous tissue plasminogen activator (t-PA) was administered (Figure 1B). After performing digital subtraction angiography (DSA) for intra-arterial thrombectomy, the right MCA showed recanalization (Figure 1C). Diffusion-weighted magnetic resonance imaging (MRI) of the brain performed on the day following admission showed diffuse infarction in the right MCA territory, and the susceptibility-weighted image (SWI) revealed hemorrhagic transformation (HT) of the right lenticulostriate arterial (LSA) territory (Figure 1D,E). On the day after admission, all other neurological symptoms had recovered, except for left hemiparesis.

Patients with atrial fibrillation and cerebral embolism are relatively more unstable than those with atherosclerotic plaques and are at a high HT risk due to reperfusion injury [1]. Therefore, the clot often moves distally from the initial occlusion site or the artery reopens due to spontaneous lysis. In addition, a sufficient number of collateral vessels in acute ischemic stroke can maintain the ischemic penumbra and prevent progression to the ischemic core, while an insufficient number of collateral vessels can increase the HT risk after therapeutic recanalization [2]. In this patient, brain CT angiography at the time of admission showed obstruction of the distal MCA. Brain MRI performed the next day showed HT using an SWI sequence in the LSA territory of the proximal site rather than the occlusion site. This phenomenon suggests that the cardiac emboli caused by atrial fibrillation blocked the proximal portion of the MCA and occluded the entrance of the LSA initially. Following this, a spontaneous process of endogenous clot lysis occurred, and the emboli migrated to the distal MCA site [3]. Therefore, distal MCA occlusion may have been observed on brain CT angiography at the time of admission. When the occluded MCA recanalizes spontaneously, the LSA territory, which does not contain sufficient number of collateral vessels, usually suffers reperfusion injury. Recanalization of the distal MCA by intravenous t-PA administration causes further progression of reperfusion injury and HT. In contrast, the distal MCA site, which contains a sufficient number of collateral vessels, could retain the ischemic penumbra when recanalization was performed by t-PA administration, suggesting that the reperfusion injury was not severe (Figure 2) [2,3].

HT in the basal ganglia is likely to cause parenchymal hematoma; hence, blood pressure should be monitored in advance. When the intracranial artery is recanalized, reperfusion injury may occur in small perforating arteries. Thus, if there is spontaneous recanalization of the distal M1 occlusion, the possibility of HT in the M1 proximal segment should be considered. Of course, each case has different patient conditions (e.g., occlusion site, duration of vessel occlusion, or collateral status), so it is recommended to refer to the conclusion of this case when treating similar patients.

## Figures and Tables

**Figure 1 diagnostics-12-00398-f001:**
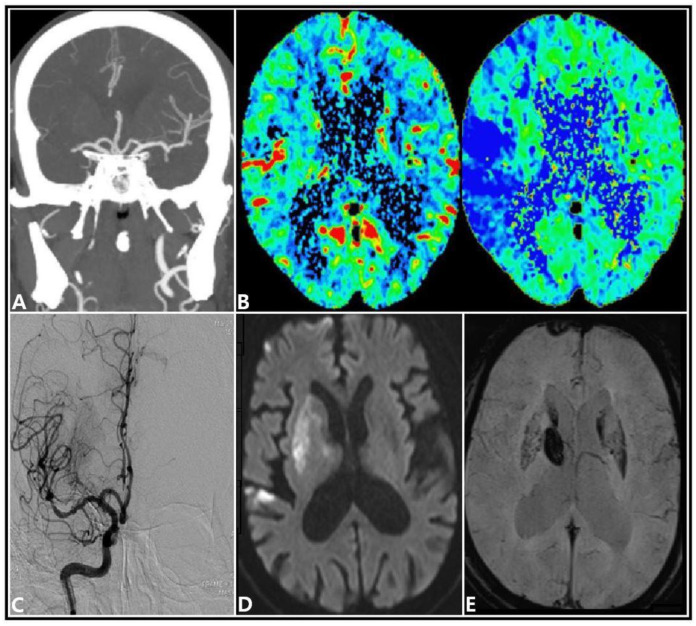
Radiologic findings of the patient. (**A**) Brain computed tomography angiography revealed distal occlusion of the right middle cerebral artery. (**B**) Severe cerebral perfusion mismatch observed in the right hemisphere. (**C**) Digital subtraction angiography showed recanalization of the right middle cerebral artery. (**D**) Brain magnetic resonance imaging showed ischemic infarction in the right cerebral hemisphere, including the lenticulostriate artery territory. (**E**) Susceptibility-weighted imaging revealed hemorrhagic transformation in the lenticulostriate artery territory.

**Figure 2 diagnostics-12-00398-f002:**
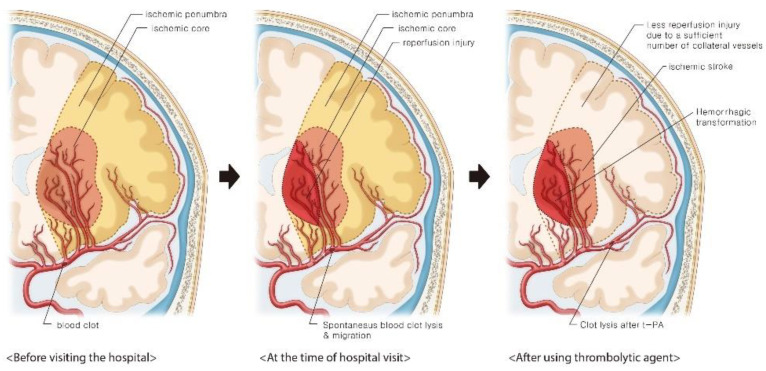
Illustration of the hemorrhagic transformation after intravenous thrombolysis.

## Data Availability

Data available on request due to restrictions, e.g., privacy or ethical.

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
