# Peer review of "Hemorrhagic Transformation after Intravenous Tissue Plasminogen Activator Administration in Acute Distal Middle Cerebral Artery Occlusion"

_diagnostics, 2022, doi:10.3390/diagnostics12020398_

Round 1
Reviewer 1 Report
This is a single-case report of hemorrhagic transformation following t-PA therapy in a patient with a cortical stroke caused by embolization. There is nothing novel in this single-case report. A large number of papers have been published on the hemorrhagic transformation following embolic stroke caused by atrial fibrillation and other cardiogenic causes with or without thrombolytic therapy. Thus, there is nothing novel in this single-case report.
Author Response
Re: Your Submission “diagnostics-1455801”
Dear editor
Thank you for your kind and positive response in regards to the manuscript that we submitted for publication in Diagnostics. We thank you for handling our manuscript and thank the reviewers for their time and effort. The evaluations and positive comments from the reviewers were very valuable in the revision of this manuscript.
We agree with all comments from Reviewers and have revised the manuscript according to their suggestions as well as the editor’s guidelines. Our revised submission for publication includes the revised main text and uploaded the MS-Word version of the main text with two figures according to the quality standards set by the formatting guidelines.
Summarized below are our detailed replies to the reviewers and a description of how we integrated them into the revised manuscript.
Reviewer 1
This is a single-case report of hemorrhagic transformation following t-PA therapy in a patient with a cortical stroke caused by embolization. There is nothing novel in this single-case report. A large number of papers have been published on the hemorrhagic transformation following embolic stroke caused by atrial fibrillation and other cardiogenic causes with or without thrombolytic therapy. Thus, there is nothing novel in this single-case report.
Author’s response: It is already well known that embolus which blocked the blood vessels can be spontaneously and easily dissolved by using intravenous t-PA, but hemorrhagic transformation (HT) is likely to occur due to reperfusion injury. However, this case study paid attention to the fact that when M1 occlusion was spontaneously recanalized due to intravenous t-PA, it was highly likely to cause hemorrhagic transformation in the basal ganglia (BG). It is a meaningful finding in the following aspects. In other words, it is possible to predict the possibility that HT has occurred in the BG. As shown in this case, it is often not able to image the brain MR quickly due to the patient's conditions (e.g., problems in the system of the medical institution and patient's own issues) even though spontaneous recanalization is confirmed through brain CT angiography and digital subtraction angiography. Moreover, even if the hemorrhagic transformation type 1 (HT-1) of ECASS II classification occurred in the infarct area of the patient, it would be difficult to confirm it on the brain CT. In this case, it can be only confirmed for certain by taking brain MRI susceptibility-weighted imaging (SWI) or gradient echo image (GRE). HI-1 poses a risk of aggravating it into a parenchymal hematoma type at any time if the patient's blood pressure is not controlled well. Therefore, from this case, we can predict that the BG is vulnerable to HT even before subsequent brain MR imaging. Consequently, clinicians can pay more attention to blood pressure fluctuations in advance, and these measures are directly related to the patient's prognosis.
In summary, we have accounted for the comments of the reviewers and believe that the manuscript is now ready for publication in Diagnostics.
Thank you and best wishes,
Sincerely,
Hyun Goo Kang, M.D. Ph.D.
Reviewer 2 Report
This study provides a case illustrating the risk factors of HT after dMCAO recanalization. The imaging data are well presented and supported by clear explanations rationalizing the authors’ conclusion. The manuscript is acceptable for publication.
Author Response
Re: Your Submission “diagnostics-1455801”
Dear editor
Thank you for your kind and positive response in regards to the manuscript that we submitted for publication in Diagnostics. We thank you for handling our manuscript and thank the reviewers for their time and effort. The evaluations and positive comments from the reviewers were very valuable in the revision of this manuscript.
We agree with all comments from Reviewers and have revised the manuscript according to their suggestions as well as the editor’s guidelines. Our revised submission for publication includes the revised main text and uploaded the MS-Word version of the main text with two figures according to the quality standards set by the formatting guidelines.
Summarized below are our detailed replies to the reviewers and a description of how we integrated them into the revised manuscript.
Reviewer 2
This study provides a case illustrating the risk factors of HT after dMCAO recanalization. The imaging data are well presented and supported by clear explanations rationalizing the authors’ conclusion. The manuscript is acceptable for publication.
Author’s response: Thank you for your comments.
In summary, we have accounted for the comments of the reviewers and believe that the manuscript is now ready for publication in Diagnostics.
Thank you and best wishes,
Sincerely,
Hyun Goo Kang, M.D. Ph.D.
Reviewer 3 Report
The authors present a well written case study demonstrating the relationship between collateralization, reperfusion, reperfusion injury, and hemorrhagic transformation. One comment are the conclusions are limited to this particular case study and not necessarily applicable to patients that present with similar imaging profiles as time, clinical presentation, acute intervention details will modify the results. The authors should consider a statement(s) about this potential limitation.
Author Response
Re: Your Submission “diagnostics-1455801”
Dear editor
Thank you for your kind and positive response in regards to the manuscript that we submitted for publication in Diagnostics. We thank you for handling our manuscript and thank the reviewers for their time and effort. The evaluations and positive comments from the reviewers were very valuable in the revision of this manuscript.
We agree with all comments from Reviewers and have revised the manuscript according to their suggestions as well as the editor’s guidelines. Our revised submission for publication includes the revised main text and uploaded the MS-Word version of the main text with two figures according to the quality standards set by the formatting guidelines.
Summarized below are our detailed replies to the reviewers and a description of how we integrated them into the revised manuscript.
Reviewer 3
The authors present a well written case study demonstrating the relationship between collateralization, reperfusion, reperfusion injury, and hemorrhagic transformation. One comment are the conclusions are limited to this particular case study and not necessarily applicable to patients that present with similar imaging profiles as time, clinical presentation, acute intervention details will modify the results. The authors should consider a statement(s) about this potential limitation.
Author’s response: We revised the conclusion section based on the reviewer's advice.
In summary, we have accounted for the comments of the reviewers and believe that the manuscript is now ready for publication in Diagnostics.
Thank you and best wishes,
Sincerely,
Hyun Goo Kang, M.D. Ph.D.
Reviewer 4 Report
Thank you for the possibility to read this very elegant case report. The images are also interesting and the aspect presented here, concerning hemorrhagic transformation in patients with acute ischemic stroke, is important in terms of clinical practice. Therefore, I think that this case report deserves publication. I have only two minor comments below.
- Please, rewrite the last sentence of the abstract to be more understandable for the reader.
- Please, correct the figure legends in the Figure 2 (in the image on the right, please write “ischemic” and “hemorrhagic” instead of “ischmic” and “hemorchagic”).
Author Response
Re: Your Submission “diagnostics-1455801”
Dear editor
Thank you for your kind and positive response in regards to the manuscript that we submitted for publication in Diagnostics. We thank you for handling our manuscript and thank the reviewers for their time and effort. The evaluations and positive comments from the reviewers were very valuable in the revision of this manuscript.
We agree with all comments from Reviewers and have revised the manuscript according to their suggestions as well as the editor’s guidelines. Our revised submission for publication includes the revised main text and uploaded the MS-Word version of the main text with two figures according to the quality standards set by the formatting guidelines.
Summarized below are our detailed replies to the reviewers and a description of how we integrated them into the revised manuscript.
Reviewer 4
Thank you for the possibility to read this very elegant case report. The images are also interesting and the aspect presented here, concerning hemorrhagic transformation in patients with acute ischemic stroke, is important in terms of clinical practice. Therefore, I think that this case report deserves publication. I have only two minor comments below.
1) Please, rewrite the last sentence of the abstract to be more understandable for the reader.
Author’s response: We revised the abstract based on the reviewer's advice.
2) Please, correct the figure legends in the Figure 2 (in the image on the right, please write “ischemic” and “hemorrhagic” instead of “ischmic” and “hemorchagic”).
Author’s response: We revised the legends in the Figure 2 based on the reviewer's advice. Thank you for your precise review.
In summary, we have accounted for the comments of the reviewers and believe that the manuscript is now ready for publication in Diagnostics.
Thank you and best wishes,
Sincerely,
Hyun Goo Kang, M.D. Ph.D.